# Relationship between the Intake of n-3 Polyunsaturated Fatty Acids and Depressive Symptoms in Elderly Japanese People: Differences According to Sex and Weight Status

**DOI:** 10.3390/nu11040775

**Published:** 2019-04-03

**Authors:** Hiromasa Tsujiguchi, Thao Thi Thu Nguyen, Daisuke Goto, Sakae Miyagi, Yasuhiro Kambayashi, Akinori Hara, Yohei Yamada, Haruki Nakamura, Yukari Shimizu, Daisuke Hori, Fumihiko Suzuki, Koichiro Hayashi, Satoko Tamai, Hiroyuki Nakamura

**Affiliations:** 1Department of Environmental and Preventive Medicine, Graduate School of Medical Science, Kanazawa University, 13-1 Takaramachi, Kanazawa, Ishikawa 920-8640, Japan; toi_fs@yahoo.com (T.T.T.N.); daigto211@gmail.com (D.G.); smiyagi@staff.kanazawa-u.ac.jp (S.M.); ykamba@med.kanazawa-u.ac.jp (Y.K.); ahara@m-kanazawa.jp (A.H.); yamada503597@gmail.com (Y.Y.); haruki_nakamura@yahoo.co.jp (H.N.); h_zu@me.com (Y.S.); fumi@dental.email.ne.jp (F.S.); k-hayashi@stu.kanazawa-u.ac.jp (K.H.); stamai411@stu.kanazawa-u.ac.jp (S.T.); hiro-n@po.incl.ne.jp (H.N.); 2Occupational and Aerospace Psychiatry Group, Graduate School of Comprehensive Human Sciences, University of Tsukuba, 1-1-1 Tennodai, Tsukuba, Ibaraki 305-8575, Japan; hori_d@mbr.nifty.com

**Keywords:** n-3 PUFA, depressive, sex, weight status, Japanese

## Abstract

n-3 polyunsaturated fatty acids (PUFAs) have been shown to have preventive effects against depression. In this study, we aimed to investigate the associations between the intake of n-3 PUFAs and depression among people according to sex and weight status. We utilized cross-sectional data from the Shika study in Japan. The study was conducted between 2013 and 2016. Data were collected from adults older than 65 years. Invitation letters were distributed to 2677 individuals, 2470 of whom participated in the study (92.3%). We assessed depressive states using the Japanese short version of the Geriatric Depression Scale (GDS-15). We assessed the intake of n-3 PUFAs using the validated food frequency questionnaire. One thousand six hundred thirty-three participants provided data, among which 327 (20.0%) exhibited depressive symptoms. When we performed the stratified analysis by sex and weight status, there were significant inverse relationships between total n-3 PUFAs, individual n-3 PUFAs, and n-3/n-6 PUFAs ratio and depressive symptoms in overweight/obese females. No correlations were observed between n-3 PUFAs intake and depressive states in males. The results demonstrated a relationship between n-3 PUFAs deficiencies and depressive states, particularly in overweight/obese females. Dietary modifications may help to prevent depressive symptoms in overweight/obese females.

## 1. Introduction

Depression is one of the major illnesses in societies worldwide and affects personal well-being, the ability to work, and the use of healthcare resources [1]. The diagnosis of major depressive disorder (DSM-V) is given if an individual has experienced five or more of the identifying symptoms for a period of at least two weeks. Of these symptoms, at least one must be “a depressed mood” or “markedly diminished interest or pleasure”. Other identifying symptoms include “significant weight loss”, “insomnia or hypersomnia”, “psychomotor agitation or retardation”, “fatigue or loss of energy”, “feelings of worthlessness or excessive or inappropriate guilt”, “diminished ability to think or concentrate, or indecisiveness”, and “recurrent thoughts of death, recurrent suicidal ideation without a specific plan, or a suicide attempt or a specific plan for committing suicide” [2]. An estimated 322 million individuals have depression worldwide [3]. A study on people aged 40 years or over in Japan reported 4.3% of males and 6.3% of females had severe depressive symptoms [4]. Depression is associated with significant disability [5]. Moreover, the economic cost of depression is a major issue for public health [6].

Depression occurs for various reasons and under different conditions. Nutrition, particularly polyunsaturated fatty acids (PUFAs), may influence depression. PUFAs are fatty acids that contain two or more unsaturated carbon–carbon double bonds with hydrogen atoms [7]. PUFAs are categorized as n-3 and n-6 fatty acids, based on their chemical structures. n-3 PUFAs include eicosapentaenoic acid (EPA), docosahexaenoic acid (DHA), docosapentaenoic acid (DPA), alpha-linolenic acid (ALA), and eicosatetraenoic acid (ETA) [7]. Longer-chain n-3 fatty acids (EPA and DHA) are synthesized by shorter-chain n-3 fatty acids (ALA) [7,8]. However, biological conversion is inefficient in humans [8,9]. In addition, shorter-chain fatty acids cannot be synthesized by humans [10,11]. Therefore, diet is an important source of these fatty acids [12]. 

n-3 PUFAs have been hypothesized to have preventive effects against depression. Previous studies have investigated the relationship between n-3 PUFAs dietary intake and depression. However, the findings obtained were conflicting [13]. Observational studies conducted on general populations are important from a prevention perspective.

Previous epidemiological studies have shown that the prevalence rate of depression is about twice in females compared to males [14]. In addition, a large number of epidemiological studies and meta-analyses have confirmed obesity to be associated with depression [15]. In recent decades, the consumption of n-3 PUFAs has been reduced and replaced by saturated fats from domestic animals and n-6 PUFAs from common vegetable oils and other sources, which changed dietary intakes of n-3 and n-6 PUFAs dramatically [16]. The higher prevalence rate of depression in females or obese people could be attributed to a vulnerability to n-3 PUFAs deficiency in these people. However, only a few researchers have evaluated the influences of sex or weight status on the relationship between n-3 PUFAs intake and depression. Understanding the influences of sex and weight status on the relationship between n-3 PUFAs intake and depression will be informative to address the target population and to maximize the beneficial effect so as to prevent depression.

Therefore, in this study, we aimed to investigate the associations between the intake of n-3 PUFAs and depression among people according to sex and weight status.

## 2. Materials and Methods

### 2.1. Study Population

We utilized cross-sectional data from the Shika study. The Shika study is an ongoing population-based survey that aims to develop advanced preventive methods for lifestyle-related diseases. It includes interviews, self-administered questionnaires, and comprehensive health examinations. Shika town is located in a rural area of the Ishikawa prefecture, Japan. The town has more than 20,000 residents [17]. Major industries include electronic component manufacturing, retail, and medical and welfare services [18]. The climate is humid subtropical. We selected four model districts in Shika town for the present study, which was conducted between October 2013 and December 2016. Data were collected from adults older than 65 years in the model districts. All adults older than 65 years who lived in these districts were eligible to participate in the present study. Invitation letters to participate were distributed to 2677 individuals, 2470 of whom participated in the study (92.3%). All subjects gave their written informed consent for inclusion before they participated in the study. The study was conducted in accordance with the Declaration of Helsinki, and the protocol was approved by the Ethics Committee of Kanazawa University (No.1491).

### 2.2. Depressive States

We assessed depressive states using the Japanese short version of the Geriatric Depression Scale (GDS-15), which consists of 15 questions developed for self-administrative surveys [19]. Higher scores indicate greater depressive symptoms [19]. A study that evaluated the validity of the Japanese version of GDS-15 recommended a cut-off score of 6/7 [20]. We used a cut-off point of 7, with scores ≥7 indicating depressive symptoms. We included participants who answered more than 12 out of the 15 questions in the analysis.

### 2.3. Nutrient Assessment

We assessed the intake of n-3 PUFAs: total n-3 PUFAs, EPA+DHA, individual PUFAs (EPA, DHA, ALA, DPA, and ETA), n-3/n-6 PUFAs ratio, and n-6 PUFAs from the validated food frequency questionnaire. We used the brief-type self-administered dietary questionnaire (BDHQ) [21,22,23]. BDHQ is based on a comprehensive version of a validated self-administered questionnaire (i.e., DHQ) [24]. BDHQ asks about dietary history in the preceding month. BDHQ lists 58 food items [22,23]. The intake of n-3 PUFAs was estimated using a computer algorithm for BDHQ [25,26]. Participants who reported an energy intake per day of less than 600 kcal/day (half of the required energy for the lowest physical activity category) or more than 4000 kcal/day (1.5 times the energy intake required for the highest physical activity category) were excluded from the analyses because they were either extremely low or extremely high energy intakes [27]. The n-3 PUFAs intake was expressed as a percentage of the total energy intake.

### 2.4. Other Variables

The weight status was based on body mass index (BMI), calculated using self-reported heights and weights. BMI was computed based on the standard formula (kg/m^2^) [28]. It was used to classify participants as underweight (<18.50), normal-weight (18.50–24.99), overweight (≥25.00), or obese (≥30.00), based on the World Health Organization reference [28]. We categorized them as underweight/normal-weight (<25.00) or overweight/obese (≥25.00). Covariates included in the multivariable analysis consisted of age, energy intake (kcal/day), carbohydrate intake (% energy), education (years of going to school), social activity (doing work or volunteering), living status (living alone or with someone), smoking status (current smoker, ex-smoker, or non-smoker), alcohol drinking, and history of chronic diseases (hypertension, stroke, myocardial infarction, diabetes, and hyperlipidemia). In the present study, participants who reported a history of stroke and/or myocardial infarction also had hypertension and/or diabetes. Therefore, we focused on hypertension, diabetes, and hyperlipidemia as confounders.

### 2.5. Statistical Analysis

Descriptive statistics were used to describe participant characteristics, depressive states, and n-3 PUFAs intake. Continuous variables were summarized as means and standard deviation (SD). Categorical variables were presented as numbers (N) and percentages (%). Differences in characteristics, depressive states, and n-3 PUFAs intake between sexes were assessed using the Student’s *t*-test (continuous variables) and the χ^2^ test (categorical variables). Differences in characteristics and n-3 PUFAs intake between participants with and without depressive symptoms were assessed using the Student’s *t*-test and the χ^2^ test. Differences in the intakes of n-3 PUFAs among participants were also assessed using a two-way analysis of variance (two-way ANOVA) with depressive states and sex as the main factors. Both depressive states and sex were between factors. Tests for interactions of depressive states and sex were done to assess sex differences. We conducted post hoc tests in order to examine the differences in the n-3 PUFAs intakes between participants with and without depressive symptoms in each sex. The two-way ANOVA was also used with depressive states and weight status as the main factors. We also tested for interactions of depressive states and weight status. Post hoc tests in each weight status category were conducted. Participants were stratified by sex and weight status for multivariate logistic regression analyses. The dependent variable was the existence of depressive symptoms. The independent variables were the intakes of each n-3 PUFAs, the n-3/n-6 PUFAs ratio, and n-6 PUFAs. Multivariate logistic regression analyses were adjusted by age, energy intake, carbohydrate intake, education, social activity, living status, smoking status, alcohol drinking, and history of chronic diseases. The odds ratios (ORs) and 95% confidence intervals (CIs) of each n-3 PUFAs for depressive states were calculated. A type I error of 0.05 was used for all analyses, with *P* values between 0.05 and 0.10 considered as borderline significance. The Statistical Package for Social Science (SPSS) for MS Windows, version 23.0 (SPSS, Inc., New York, NY, USA) was used for statistical analyses.

## 3. Results

### 3.1. Participant Characteristics

The characteristics of the studied population are shown in Table 1. A total of 1633 participants provided data for the variables of interest and were included in the analyses. The number of males was 720, while that of females was 913. The average ages of the participants were 73.5 (SD = 6.8) years for males and 75.2 (SD = 7.8) years for females. A total of 327 (20.0%) of participants had depressive symptoms. Among them, 147 (20.4%) were male and 180 (19.7%) were female. Females had significantly higher total n-3 PUFAs, ALA, and n-6 PUFAs intakes than males. The dietary sources of n-3 PUFAs in the studied population are shown in Appendix A.

### 3.2. Participant Characteristics and n-3 PUFAs Intake According to Depressive States

Participant characteristics and n-3 PUFAs intake according to depressive states are shown in Table 2. Participants with depressive symptoms were significantly older than those without. Participants with depressive symptoms had a significantly lower energy intake, less education, less social activity, less household size, less alcohol drinking than those without. The levels of seven n-3 PUFAs were significantly lower in participants with depressive symptoms than in those without. Although no significant difference was observed between the two groups, the n-3/n-6 PUFAs ratio and the n-6 PUFAs were slightly lower in participants with depressive symptoms.

### 3.3. Influence of Sex on the Association between n-3 PUFAs Intake and Depressive States

The influence of sex on the association between n-3 PUFAs intake and depressive states is shown in Table 3. There was a significant or borderline significant main effect of depressive states in seven n-3 PUFAs, excluding n-3/n-6 PUFAs ratio and n-6 PUFAs. The tests for interactions of sex and depressive states were statistically significant or tended to be significant in seven n-3 PUFAs and n-6 PUFAs, excluding n-3/n-6 PUFAs ratio, which indicated that the associations between n-3 PUFAs intake and depressive states differed by sex. Post hoc tests showed, among females, the levels of all seven n-3 PUFAs, n-3/n-6 PUFAs ratio, and n-6 PUFAs were significantly lower in participants with depressive states than in those without. In contrast, in males, none of the n-3 PUFAs, n-3/n-6 PUFAs ratio, or n-6 PUFAs levels were significantly lower in participants with depressive symptoms than in those without.

### 3.4. Influence of Weight Status on the Association between n-3 PUFAs Intake and Depressive States

The influence of weight status on the association between n-3 PUFAs intake and depressive states is shown in Table 4. The main effects of depressive states were significant or tended to be significant in all n-3 PUFAs, the n-3/n-6 PUFAs ratio, and n-6 PUFAs. The interactions of weight status and depressive states tended to be statistically significant in EPA+DHA, EPA, DHA, and DPA. When we performed post hoc analyses among overweight/obese participants, the levels of seven n-3 PUFAs and the n-3/n-6 PUFAs ratio were significantly lower or tended to be significantly lower in participants with depressive symptoms than without depressive symptoms.

### 3.5. Relationships between n-3 PUFAs Intake and Depressive States by Sex and Weight Status

The relationships between n-3 PUFAs intake and depressive states by sex and weight status are shown in Table 5 and Figure 1. When we conducted stratified analyses by sex and weight status, there were significant inverse relationships for total n-3 PUFAs, EPA+DHA, EPA, DHA, DPA, ETA, and n-3/n-6 PUFAs ratio in overweight/obese females. The intakes of total n-3 PUFAs and ALA were also inversely related to depressive symptoms among underweight/normal-weight females. No correlations were observed between n-3 PUFAs intake or n-3/n-6 PUFAs ratio and depressive states in males.

## 4. Discussion

The results of the present study showed that a higher intake of n-3 PUFAs was associated with a lower risk of depressive symptoms, especially in overweight/obese females. To the best of our knowledge, this is the first study to demonstrate differences according to sex and weight status on the relationship between n-3 PUFAs intake and depressive symptoms.

The present results are consistent with previous findings, showing that n-3 PUFAs intake is inversely associated with depression [29]. According to a systematic review, most studies found that n-3 PUFAs intake was inversely associated with depression, though some studies found no association [29]. The difference observed between these findings may be due to the target population examined (middle-aged vs. elderly individuals) or the study design. In addition, most of the studies that investigated the relationship between n-3 PUFAs intake and depression were not stratified by sex or weight status. This may limit appropriate comparisons with the present results, which were stratified by sex and weight status.

Regarding participant characteristics, a relationship was found in females but not in males. This could be due to the difference in sample size between the males and females in our study (*n* = 720 (males) and *n* = 913 (females)). Meanwhile, several epidemiological studies have found inverse associations between n-3 PUFAs and depressive symptoms in females but not in males [30,31,32,33]. Colangelo et al. [30] reported that higher total n-3 PUFAs, EPA+DHA, EPA, and DHA intakes predicted less severe depressive symptoms in females only. Beydoun et al. [31] also found that n-3 PUFAs intake and the n-3/n-6 PUFAs ratio were inversely associated with depressive symptoms in females. Persons et al. [33] found a positive association of dietary DHA+EPA with baseline depressive symptoms, though they found no associations at follow-up. In the study of Lucas et al. [33], although an inverse association of α-linolenic acid with clinical depression was found, there was no association for total n-3 PUFAs. They also found that depression was positively linked to the n-3/n-6 PUFAs ratio. 

Sex differences in the association between the intake of n-3 PUFAs and depressive symptoms can be explained by some mechanisms. Depression is considered to be secondary to inflammatory disorders. For example, inflammatory proteins, such as C-reactive protein (CRP) [34], and cytokines, such as interleukin (IL)-6 [35] and tumor necrosis factor (TNF)-α [36], have been elevated in people with depression. Pro-inflammatory cytokines alter serotonin metabolism, reduce synaptic plasticity, and increase the risk of developing depression [37,38,39]. The anti-inflammatory effects of n-3 PUFAs have been considered to play an important role in counteracting the inflammatory processes occurring in depression [40]. One possible explanation is that plasma DHA levels are higher in females than in males independent of DHA intake due to sex hormones [41,42]. It has been shown that estrogens may contribute to higher DHA concentrations [41,42].

It has not been clarified whether n-3 PUFAs intake is associated with depressive states among overweight/obese people. We could not find other epidemiological studies which focused on the modification by weight status on the association between n-3 PUFAs intake and depressive symptoms. It is suggested that obesity is also a disorder of inflammation [43]. Therefore, inflammation could link obesity with depression [44]. Hence, n-3 PUFAs could play a role in depressive symptoms via inflammatory processes, especially in the context of being overweight or obese.

Regarding ALA, a correlation was not observed in overweight/obese people. This could be due to a high n-6 PUFAs intake in overweight/obese people. Some studies suggest that overweight/obese people consume a diet with a low n-3/n-6 ratio [45]. It is reported that very little ALA is converted to EPA and DHA in the presence of high levels of n-6 PUFAs [46,47]. 

Furthermore, estrogen has been shown to increase inflammation [48,49]. Females have a stronger immune response than males [49,50]. The observed results in overweight/obese females could be due to a stronger immune response in females and the anti-inflammatory effects of n-3 PUFAs compared to males in all weight categories and underweight/normal-weight females.

A cross-national ecological study reported that an intake of 0.750 g/day of n-3 PUFAs was sufficient to protect 98% of the population from the risk of depression [51]. The World Health Organization (WHO) recommended an intake of 0.250 g/day of EPA plus DHA for non-pregnant/non-lactating adult females [52]. Dietary sources of longer-chain n-3 PUFAs are fish and wild animals (e.g., fatty fish, white fish, shellfish, other sea foods, eggs, and wild game) [11,53,54,55]. Dietary sources of shorter-chain n-3 PUFAs are plants (e.g., nuts, seeds, flaxseed, and rapeseed oil) [11,53,54,55].

Our study had several limitations. The study design was cross-sectional; therefore, the causality of relationships cannot be assessed. In addition, we obtained data from a subsample of a larger cohort. This may have introduced a selection bias, because these subsamples may not resemble the target population. Furthermore, despite being validated, the depression scale only suggests the presence of a depressive symptom. Another limitation is that self-reported assessment methods were subject to recall bias or misreporting. Overweight or obese individuals are more likely to misreport information [56]. However, the instruments used in the present study were validated, and evidence suggests that errors by misreporting may be cancelled out, at least in part, by energy adjustments [57,58].

## 5. Conclusions

The results of the present study demonstrated a relationship between n-3 PUFAs deficiencies and depressive states, particularly in overweight/obese females. Our findings suggest that dietary modifications may help to prevent depressive symptoms in overweight/obese females, informing how interventions can effectively target specific populations for prevention. Further research is needed to clarify whether this relationship is consistent.

## Figures and Tables

**Figure 1 nutrients-11-00775-f001:**
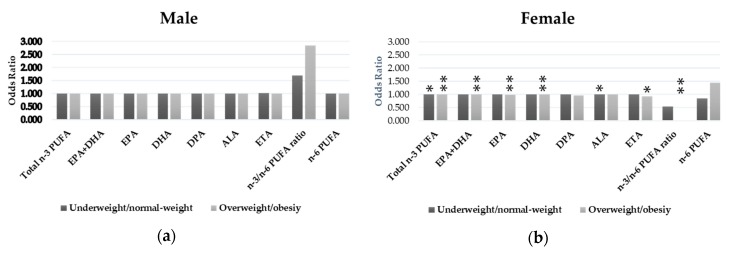
(**a**) Relationships between n-3 PUFAs intake and depressive states in males by weight status (* *p*-values under 0.05, ** *p*-values under 0.01), and (**b**) relationships between n-3 PUFAs intake and depressive states in females by weight status (* *p*-values under 0.05, ** *p*-values under 0.01).

**Table 1 nutrients-11-00775-t001:** Participant characteristics.

	Total (*N* = 1633)		Male (*N* = 720)		Female (*N* = 913)		*p* *
	Ave (*n*)	SD (%)	Ave (*n*)	SD (%)	Ave (*n*)	SD (%)	
Age	74.5	7.4	73.5	6.8	75.2	7.8	**<0.001**
BMI	22.9	3.2	23.3	3	22.6	3.3	**<0.001**
Energy (kcal)	1824.1	610.9	2014.5	632.3	1673.9	549.1	**<0.001**
Carbohydrate (% energy)	55.4	8.5	54.0	8.9	56.5	8.1	**<0.001**
Education	10.4	2.4	10.7	2.5	10.1	2.2	**<0.001**
Social activity (none)	1237	78.3%	523	75.0%	714	80.9%	**0.005**
Household size (being single)	207	12.9%	56	7.9%	151	16.9%	**<0.001**
Drinking	905	58.5%	526	75.4%	379	44.6%	**<0.001**
Smoking	165	10.4%	146	20.7%	19	2.1%	**<0.001**
Hypertension	643	39.4%	310	43.1%	333	36.5%	**0.007**
Diabetes	229	14.0%	122	16.9%	107	11.7%	**0.003**
Hyperlipidemia	314	19.2%	102	14.2%	212	23.2%	**<0.001**
Depressive state	327	20.0%	147	20.4%	180	19.7%	0.725
Total n-3 PUFAs (% energy)	1.382	0.481	1.338	0.470	1.417	0.487	**0.001**
EPA+DHA (% energy)	0.539	0.315	0.531	0.310	0.545	0.320	0.368
EPA (% energy)	0.204	0.127	0.201	0.124	0.207	0.129	0.390
DHA (% energy)	0.335	0.189	0.330	0.186	0.339	0.192	0.354
DPA (% energy)	0.058	0.033	0.057	0.033	0.059	0.034	0.417
ALA (% energy)	0.714	0.224	0.679	0.211	0.741	0.230	**<0.001**
ETA (% energy)	0.020	0.013	0.020	0.012	0.020	0.013	0.551
n-3/n-6 PUFAs ratio	0.305	0.107	0.307	0.105	0.302	0.108	0.343
n-6 PUFAs	5.149	1.339	4.928	1.265	5.323	1.370	**<0.001**

* *t*-tests for continuous variables and chi-square tests for categorical variables by sex (*p*-values under 0.05 are highlighted in bold).

**Table 2 nutrients-11-00775-t002:** Participant characteristics and n-3 polyunsaturated fatty acids (PUFAs) intake according to depressive states (*t*-tests and chi-square tests).

	Without Depressive States (*N* = 1306)		With Depressive States (*N* = 327)		*p* *
	Ave (*n*)	SD (%)	Ave (*n*)	SD (%)	
Age	73.8	7.1	77.3	8.1	**<0.001**
BMI	23	3.1	22.6	3.5	0.064
Energy (kcal)	1852.2	618.2	1712	568	**<0.001**
Carbohydrate (% energy)	55.1	8.5	56.6	8.5	**0.004**
Education	10.5	2.4	9.9	2.4	**<0.001**
Social activity (none)	962	76.0%	275	87.6%	**<0.001**
Household size (being single)	159	12.4%	48	15.0%	0.206
Drinking	746	60.3%	159	51.1%	0.003
Smoking	138	10.8%	27	8.7%	0.264
Hypertension	511	39.1%	132	40.4%	0.682
Diabetes	180	13.8%	49	15.0%	0.576
Hyperlipidemia	258	19.8%	56	17.1%	0.278
Total n-3 PUFAs (% energy)	1.402	0.487	1.303	0.450	**<0.001**
EPA+DHA (% energy)	0.548	0.321	0.504	0.290	**0.017**
EPA (% energy)	0.208	0.129	0.189	0.117	**0.011**
DHA (% energy)	0.340	0.192	0.315	0.174	**0.033**
DPA (% energy)	0.059	0.034	0.055	0.030	**0.032**
ALA (% energy)	0.722	0.224	0.679	0.220	**0.002**
ETA (% energy)	0.021	0.013	0.019	0.011	**0.028**
n-3/n-6 PUFAs ratio	0.307	0.109	0.295	0.100	0.082
n-6 PUFAs	5.179	1.332	5.026	1.363	0.063

* *t*-tests for continuous variables and chi-square tests for categorical variables by depressive states (*p*-values under 0.05 are highlighted in bold).

**Table 3 nutrients-11-00775-t003:** Influence of sex on the association between n-3 PUFAs intake and depressive states (two-way ANOVA).

		Without Depressive States (*N* = 1306)			With Depressive States (*N* = 327)			*p*1 ^a^	*p*2 ^b^	*p*3 ^c^	*p*4 ^d^
		Average	95% CI		Average	95% CI					
			Lower	Upper		Lower	Upper				
Total n-3 PUFAs	Male (*N* = 720)	1.340	1.301	1.379	1.329	1.254	1.403	**0.002**	**0.008**	0.798	**<0.001**
	Female (*N* = 914)	1.450	1.415	1.486	1.281	1.217	1.346				
EPA+DHA	Male (*N* = 720)	0.531	0.505	0.556	0.533	0.484	0.582	**0.045**	**0.034**	0.937	**0.002**
	Female (*N* = 914)	0.561	0.537	0.585	0.480	0.440	0.521				
EPA	Male (*N* = 720)	0.201	0.191	0.211	0.201	0.181	0.221	**0.030**	**0.030**	0.995	**<0.001**
	Female (*N* = 914)	0.213	0.204	0.223	0.179	0.163	0.196				
DHA	Male (*N* = 720)	0.329	0.314	0.345	0.332	0.302	0.361	0.058	**0.037**	0.898	**0.003**
	Female (*N* = 914)	0.348	0.334	0.362	0.301	0.277	0.325				
DPA	Male (*N* = 720)	0.057	0.055	0.060	0.058	0.052	0.063	0.055	**0.043**	0.944	**0.003**
	Female (*N* = 914)	0.060	0.058	0.063	0.052	0.048	0.056				
ALA	Male (*N* = 720)	0.681	0.664	0.699	0.668	0.633	0.704	**0.004**	**0.047**	0.528	**<0.001**
	Female (*N* = 914)	0.754	0.738	0.771	0.687	0.654	0.720				
ETA	Male (*N* = 720)	0.020	0.019	0.021	0.020	0.018	0.022	**0.047**	0.053	0.976	**0.003**
	Female (*N* = 914)	0.021	0.020	0.022	0.018	0.016	0.020				
n-3/n-6 PUFAs	Male (*N* = 720)	0.308	0.299	0.316	0.307	0.290	0.324	0.114	0.139	0.946	**0.022**
ratio	Female (*N* = 914)	0.306	0.298	0.314	0.286	0.272	0.300				
n-6 PUFAs	Male (*N* = 720)	4.927	4.819	5.036	4.933	4.719	5.147	0.102	0.089	0.964	**0.013**
	Female (*N* = 914)	5.377	5.281	5.473	5.102	4.908	5.295				

^a^: Between with and without depressive states groups. ^b^: Interaction by depressive states groups and sex. ^c^: Bonferroni post hoc tests between males with and without depressive symptoms. ^d^: Bonferroni post hoc tests between females with and without depressive symptoms. ^a–d^: *p*-values under 0.05 are highlighted in bold.

**Table 4 nutrients-11-00775-t004:** Influence of weight status on the association between n-3 PUFAs intake and depressive states (two-way ANOVA).

		Without Depressive States (*N* = 1306)			With Depressive States (*N* = 327)			*p*1 ^a^	*p*2 ^b^	*p*3 ^cd^	*p*4 ^d^
		Average	95% CI		Average	95% CI					
			Lower	Upper		Lower	Upper				
Total n-3 PUFAs	Underweight/normal-weight (*N* = 1255)	1.392	1.362	1.422	1.318	1.260	1.377	**<0.001**	0.109	0.028	**0.003**
	Overweight/obese (*N* = 378)	1.434	1.380	1.487	1.244	1.131	1.357				
EPA+DHA	Underweight/normal-weight (*N* = 1255)	0.544	0.525	0.564	0.518	0.480	0.557	**0.005**	0.083	0.235	**0.010**
	Overweight/obese (*N* = 378)	0.559	0.524	0.594	0.451	0.376	0.525				
EPA	Underweight/normal-weight (*N* = 1255)	0.207	0.199	0.215	0.195	0.179	0.210	**0.003**	0.091	0.176	**0.009**
	Overweight/obese (*N* = 378)	0.212	0.198	0.227	0.168	0.138	0.198				
DHA	Underweight/normal-weight ( *N* = 1255)	0.338	0.326	0.349	0.323	0.300	0.346	**0.006**	0.079	0.283	**0.011**
	Overweight/obese (*N* = 378)	0.347	0.326	0.368	0.282	0.238	0.327				
DPA	Underweight/normal-weight (*N* = 1255)	0.059	0.057	0.061	0.056	0.052	0.060	**0.007**	0.094	0.260	**0.013**
	Overweight/obese (*N* = 378)	0.060	0.056	0.064	0.049	0.041	0.057				
ALA	Underweight/normal-weight (*N* = 1255)	0.717	0.703	0.731	0.677	0.649	0.704	**0.005**	0.669	0.010	0.067
	Overweight/obese (*N* = 378)	0.740	0.715	0.765	0.686	0.633	0.738				
ETA	Underweight/normal-weight (*N* = 1255)	0.020	0.020	0.021	0.019	0.018	0.021	**0.007**	0.119	0.221	**0.016**
	Overweight/obese (*N* = 378)	0.021	0.020	0.022	0.017	0.014	0.020				
n-3/n-6 PUFAs	Underweight/normal-weight (*N* = 1255)	0.307	0.300	0.313	0.300	0.287	0.313	**0.030**	0.201	0.007	**0.014**
ratio	Overweight/obese (*N* = 378)	0.307	0.295	0.319	0.280	0.254	0.305				
n-6 PUFAs	Underweight/normal-weight (*N* = 1255)	5.148	5.065	5.232	5.018	4.854	5.181	0.079	0.647	0.162	0.211
	Overweight (*N* = 378)	5.279	5.130	5.429	5.056	4.740	5.372				

^a^: Between with and without depressive states groups. ^b^: Interaction by depressive states and weight status. ^c^: Bonferroni post hoc tests between underweight/normal-weight participants with and without depressive symptoms. ^d^: Bonferroni post hoc tests between overweight/obese people with and without depressive symptoms. ^a–d^: *p*-values under 0.05 are highlighted in bold.

**Table 5 nutrients-11-00775-t005:** Relationship between n-3 PUFAs intake and depressive states by sex and weight status (logistic regression analysis).

Weight Status	PUFAs	Male				Female			
		OR	95% CI		*p*	OR	95% CI		*p* *
			Lower	Upper			Lower	Upper	
Underweight/normal-weight	Total n-3 PUFAs	1.000	1.000	1.001	0.570	0.999	0.998	1.000	**0.020**
	EPA+DHA	1.000	1.000	1.001	0.362	1.000	0.999	1.000	0.407
	EPA	1.001	0.999	1.003	0.383	0.999	0.997	1.001	0.330
	DHA	1.001	0.999	1.002	0.349	0.999	0.998	1.001	0.467
	DPA	1.005	0.996	1.013	0.317	0.998	0.990	1.005	0.544
	ALA	1.000	0.998	1.001	0.639	0.998	0.997	1.000	**0.013**
	ETA	1.011	0.988	1.034	0.364	0.994	0.975	1.014	0.550
	n-3/n-6 PUFAs ratio	1.691	0.143	20.050	0.677	0.544	0.080	3.719	0.535
	n-6 PUFAs	0.999	0.806	1.239	0.994	0.856	0.693	1.057	0.148
Overweight/obesity	Total n-3 PUFAs	1.000	0.999	1.002	0.604	0.997	0.996	0.999	**0.007**
	EPA+DHA	1.000	0.999	1.002	0.557	0.997	0.994	0.999	**0.007**
	EPA	1.001	0.997	1.005	0.547	0.991	0.985	0.997	**0.005**
	DHA	1.001	0.998	1.003	0.565	0.995	0.990	0.999	**0.009**
	DPA	1.004	0.989	1.020	0.579	0.968	0.944	0.992	**0.009**
	ALA	1.000	0.998	1.002	0.955	1.000	0.997	1.003	0.783
	ETA	1.009	0.970	1.050	0.640	0.922	0.866	0.982	**0.012**
	n-3/n-6 PUFAs ratio	2.856	0.063	130.369	0.590	0.001	0.001	0.011	**0.003**
	n-6 PUFAs	0.993	0.695	1.417	0.968	1.441	0.830	2.502	0.195

* Adjusted for age, energy, carbohydrate, education, social activity, household size, drinking, smoking, hypertension, diabetes, and hyperlipidemia (*p*-values under 0.05 are highlighted in bold).

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
