# Peer review of "Relationship between the Intake of n-3 Polyunsaturated Fatty Acids and Depressive Symptoms in Elderly Japanese People: Differences According to Sex and Weight Status"

_nutrients, 2019, doi:10.3390/nu11040775_

Round 1

Reviewer 1 Report

Thank you for the opportunity to review this paper. The information provided supports the benefits of consuming the n-3 fatty acids. In addition, the manuscript provides input on possible nutrient interventions to prevent or reduce the risk of depressive symptoms in adulthood, especially for women.

Comments and suggestions are detailed below:

Line 19 – abstract: remove hypothesized since thought the manuscript you are presenting references that support the beneficial role of the n-3 against depression.

Line 26: Spell out 1,633 since you are starting the sentence.

I think the introduction will benefit if Japan’s depression rates are included. This will give the manuscript more relevance as to the reasons to study this topic.

Lines 38-46: Too long sentence, it’s difficult to understand.

Line 55: Add “in humans” before references [7,8].

Line 73: You are missing a space between n-3 and PUFA.

Lines 105-107: How/why the cut off points of 600 and 4,000 kcal/day were selected?

Results: Much of the information in the paragraphs is already in the tables. Choose what information you want to leave in the table to avoid duplication. For example, in lines 158-161 the information is already in the table and having it in the paragraph makes it difficult for the reader to understand.  Check this for all of the tables.

Results: I think it would be worth to mention the total of the n-6 somewhere since you are mentioning the n-3/n-6 ratio.

Tables: Check the headings, some words are in bold and others are not. Simplify the tables, they are a bit crowded making it difficult for the reader to focus on the important information.  For example, in tables 3 and 4 you can remove the ‘male and females’ column and the ‘% energy’ since everything is the same.

Line 195: Remove “were” before tended.

Line 233-234: Did you find any different results if you wouldn’t have stratified by sex and weight status?

Thought the manuscript: Be consistent on how you denote the n-3/n-6 ratio.

Line 251: You are missing a space before the references.

Line 255: I don’t agree with this sentence since the conversion in humans is very limited – you are even mentioning this at the beginning of the manuscript. If you want to keep this, then I would suggest mentioning the conversion rates of females vs males.

Line 275: You are missing a space before the reference.

Line 279: Was the population consuming any of the dietary sources you are mentioning? If so, then it would be worth to mention the main n-3 dietary sources of the population (data from the food frequency questionnaire).

Line 285: remove the “s” in symptoms.

Line 286: Add “to” before misreport.

Author Response

29th May, 2019

Nutrients

Dear Ms. Poppy Fan

Manuscript ID: nutrients-476980
Type of manuscript: Article
Title: Relationship between the intake of n-3 polyunsaturated fatty acids and depressive symptoms in elderly Japanese people: differences according to sex and weight status
Authors: Hiromasa Tsujiguchi *, Thao Thi Thu Nguyen, Daisuke Goto, Sakae Miyagi, Yasuhiro Kambayashi, Akinori Hara, Yamada Yohei, Haruki Nakamura, Yukari Shimizu, Daisuke Hori, Fumihiko Suzuki, Koichiro Hayashi, Satoko Tamai, Hiroyuki Nakamura
Received: 19 March 2019

Thank you very much for your e-mail dated March 27th 2019 with the reviewers’ comments. We are returning herewith the manuscript revised according to your e-mail. We have carefully reviewed the comments and have revised the manuscript accordingly. Our responses are given in a point-by-point manner below. Changes to the manuscript are shown using the Track Changes function.

Hereafter, the comments by the reviewers are shown in bold text.

Response to reviewer #1

1)    Line 19-abstract: remove hypothesized since thought the manuscript you are presenting references that support the beneficial role of the n-3 against depression.

We agree. We removed hypothesized and changed the sentence to “n-3 polyunsaturated fatty acids (PUFA) have been shown to have preventive effects against depression” (line 19-20).

2)     Line 26: Spell out 1,633 since you are starting the sentence.

Accordingly, we have changed 1,633 to “One thousand six hundred thirty-three” (line 26).

3)     I think the introduction will benefit if Japan’s depression rates are included. This will give the manuscript more relevance as to the reasons to study this topic.

We agree. We added Japan’s depression rates (line 54-55).

4)     Lines 38-46: Too long sentence, it’s difficult to understand.

In accordance with the reviewer’s comment, we split a long sentence into three simpler sentences as below (line 38-45).

The diagnosis of major depressive disorder (DSM-V) is given if an individual has experienced five or more of the identifying symptoms for a period of at least 2 weeks. Of these symptoms, at least one must be “a depressed mood” or “markedly diminished interest or pleasure”. Other identifying symptoms include “significant weight loss”, “insomnia or hypersomnia”, “psychomotor agitation or retardation”, “fatigue or loss of energy”, “feelings of worthlessness or excessive or inappropriate guilt”, “diminished ability to think or concentrate, or indecisiveness”, and “recurrent thoughts of death, recurrent suicidal ideation without a specific plan, or a suicide attempt or a specific plan for committing suicide” [2].

5)     Line 55: Add “in humans” before references [7,8].

Accordingly, we added “in humans” before references [7,8] (line 63).

6)     Line 73: You are missing a space between n-3 and PUFA.

Accordingly, we added space between n-3 and PUFA (line 81).

7)     Lines 105-107: How/why the cut off points of 600 and 4,000 kcal/day were selected?

There is not enough evidence for underreport or overreport regarding energy intake. Dr. Satoshi Sasaki who developed BDHQ suggested to select the cut off points of 600 and 4,000 kcal/day as criteria for underreport or overreport. We followed Dr. Sasaki’s suggestions. Accordingly, we modified the manuscript (line 113-115).

8)     Results: Much of the information in the paragraphs is already in the tables. Choose what information you want to leave in the table to avoid duplication. For example, in lines 158-161 the information is already in the table and having it in the paragraph makes it difficult for the reader to understand. Check this for all of the tables.

Accordingly, we made descriptions of results easier to understand (line 160-161,166-175,183-184,186-187,190-194,205-208,211-215,225-227,229).

9)     Results: I think it would be worth to mention the total of the n-6 somewhere since you are mentioning the n-3/n-6 ratio.

Accordingly, we added results of analysis and descriptions of n-6 PUFA to Results.

10)   Tables: Check the headings, some words are in bold and others are not. Simplify the tables, they are a bit crowded making it difficult for the reader to focus on the important information. For example, in tables 3 and 4 you can remove the ‘male and females’ column and the’ % energy’ since everything is the same.

Accordingly, we modified tables.

11)   Line 195: Remove “were” before tended.

Accordingly, we removed “were” before tended (line 207).

12)   Line 233-234: Did you find any different results if you wouldn’t have stratified by sex and weight status?

We didn’t find any significant differences in results whether stratified by sex and weight status or not in this study. In this regard, however, the p-value was low and the result was clear when stratified.

13)   Throughout the manuscript: Be consistent on how you denote the n-3/n-6 ratio.

Accordingly, we changed n-6:n-3 to n-3/n-6 (Line 261) and n-6 to n-3 to n-3/n-6 (Line 285).

14)   Line 251: You are missing a space before the references.

Accordingly, we added a space before the references (line 267).

15)   Line 255; I don’t agree with this sentence since the conversion in humans is very limited-you are even mentioning this at the beginning of the manuscript. If you want to keep this, then I would suggest mentioning the conversion rates of females vs males.

We agree. We removed descriptions of it (line 271-276).

16)   Line 275: You are missing a space before the reference.

Accordingly, we added a space before the reference (line 292).

17)   Line 279: Was the population consuming any of the dietary sources you are mentioning? If so, then it would be worth to mention the main n-3 dietary sources of the population (data from the food frequency questionnaire).

The study population consumed fishes, oils and pulses as main n-3 dietary sources. We added Supplementary Table (line 312-313) and upload as the separate file.

18)   Line 285: remove the “s” in symptoms.

Accordingly, we removed the “s” in symptoms (line 302).

19)   Line 286: Add “to” before misreport.

Accordingly, we added “to” before misreport (line303).

Reviewer 2 Report

The study demonstrated that higher intake of omega-3 fatty acids was associated with reduced depressive symptoms, in particular in overweight/obese females.

The authors reported some limitations of the study. I agree with them. Anyway, the study is original and novel as it is the first study demonstrating differences according to sex and weight status on the relationship between the intake of omega-3 fatty acids and depressive symptoms.

In my opinion, the authors should better clarify (and justify) the statistical design (e.g., dependent and indepent variables in the regression analyses, between and within factors in the ANOVA) and add figures for the main results. In the tables, the significant results should be evidenced (e.g., in bold) to help readers.

Author Response

29th May, 2019

Nutrients

Dear Ms. Poppy Fan

Manuscript ID: nutrients-476980
Type of manuscript: Article
Title: Relationship between the intake of n-3 polyunsaturated fatty acids and depressive symptoms in elderly Japanese people: differences according to sex and weight status
Authors: Hiromasa Tsujiguchi *, Thao Thi Thu Nguyen, Daisuke Goto, Sakae Miyagi, Yasuhiro Kambayashi, Akinori Hara, Yamada Yohei, Haruki Nakamura, Yukari Shimizu, Daisuke Hori, Fumihiko Suzuki, Koichiro Hayashi, Satoko Tamai, Hiroyuki Nakamura
Received: 19 March 2019

Thank you very much for your e-mail dated March 27th 2019 with the reviewers’ comments. We are returning herewith the manuscript revised according to your e-mail. We have carefully reviewed the comments and have revised the manuscript accordingly. Our responses are given in a point-by-point manner below. Changes to the manuscript are shown using the Track Changes function.

Hereafter, the comments by the reviewers are shown in bold text.

Response to reviewer #2

1)     In my opinion, the authors should better clarify (and justify) the statistical design (e.g., dependent and independent variables in the regression analyses, between and within factors in the ANOVA) and add figure for the main results. In the tables, the significant results should be evidenced (e.g., in bold) to help readers.

We agree. In accordance with the reviewer’s comment, we added the description of the statistical design in 2.5.Statistical analysis. Furthermore, we created figures and added regarding Table 5 which is the main result (line 235-237). In addition, we emphasized numerical values which are noticeable results statistically in tables by using bold.

We also corrected ages (line 23,92-93) and reference numbers.

I would like to thank the editor, assistant editor and the reviewers for their helpful comments and hope that the revised manuscript is now suitable for publication in Environmental Health and Preventive Medicine.

Yours sincerely,

Hiromasa Tsujiguchi, PhD,

Assistant Professor,

Department of Environmental and Preventive Medicine, Graduate School of Medical Sciences, Kanazawa University, 13-1 Takaramachi, Kanazawa, Ishikawa, 920-8640, Japan

Phone:     +81-76-265-2288